In situ effects of simulated overfishing and eutrophication on settlement of benthic coral reef invertebrates in the Central Red Sea

Jessen Christian 1 christian.jessen@zmt-bremen.de
Voolstra Christian R. 2
Wild Christian 1 3
1 Coral Reef Ecology Group (CORE), Leibniz Center for Tropical Marine Ecology , Bremen , Germany
2 Red Sea Research Center, King Abdullah University of Science and Technology (KAUST) , Thuwal , Saudi Arabia
3 Faculty of Biology and Chemistry, University of Bremen , Germany
Ford Alex
Electronic publication date: 2014 Apr 8
Publication date: 2014
Volume: 2
Electronic Location ID: e339
Received 2014 Feb 2; Accepted 2014 Mar 21
Copyright: © 2014 Jessen et al.
Copyright year: 2014
Copyright holder: Jessen et al.
License: This is an open access article distributed under the terms of the Creative Commons Attribution License, which permits unrestricted use, distribution, reproduction and adaptation in any medium and for any purpose provided that it is properly attributed. For attribution, the original author(s), title, publication source (PeerJ) and either DOI or URL of the article must be cited.
License URL: https://creativecommons.org/licenses/by/4.0/

Keywords: Recruitment, Sessile invertebrates, Nutrient enrichment, Overfishing, Bioindicator, Settlement, Red Sea, Coral reefs, Saudi Arabia, Caging experiments

Funding: KAUST baseline funds DFG grant Wi 2677/6-1 The study was partially financed through KAUST baseline funds to CRV and partially supported by a DFG grant Wi 2677/6-1 to CW. The funders had no role in study design, data collection and analysis, decision to publish, or preparation of the manuscript.

==============================
In the Central Red Sea, healthy coral reefs meet intense coastal development, but data on the effects of related stressors for reef functioning are lacking. This in situ study therefore investigated the independent and combined effects of simulated overfishing through predator/grazer exclusion and simulated eutrophication through fertilizer addition on settlement of reef associated invertebrates on light-exposed and -shaded tiles over 4 months. At the end of the study period invertebrates had almost exclusively colonized shaded tiles. Algae were superior settling competitors on light-exposed tiles. On the shaded tiles, simulated overfishing prevented settlement of hard corals, but significantly increased settlement of polychaetes, while simulated eutrophication only significantly decreased hard coral settlement relative to controls. The combined treatment significantly increased settlement of bryozoans and bivalves compared to controls and individual manipulations, but significantly decreased polychaetes compared to simulated overfishing. These results suggest settlement of polychaetes and hard corals as potential bioindicators for overfishing and eutrophication, respectively, and settlement of bivalves and bryozoans for a combination of both. Therefore, if the investigated stressors are not controlled, phase shifts from dominance by hard corals to that by other invertebrates may occur at shaded reef locations in the Central Red Sea.

Introduction

Overfishing and eutrophication are among the most serious local stressors for coral reefs, worldwide and in the Red Sea (Burke et al., 2011). These stressors can strongly affect invertebrate settlement. Settlement (i.e., the permanent attachment to the substratum) of sessile invertebrate larvae is an irreversible process and is thus of critical importance for invertebrate life-cycles (Harrison & Wallace, 1990).

Invertebrate settlement can be influenced by numerous factors such as water flow (Mullineaux & Garland, 1993), abundance and composition of microbial biofilms (Hadfield, 2011; Sawall, Richter & Ramette, 2012; Tran & Hadfield, 2011), benthic macroalgae (Arnold, Steneck & Mumby, 2010; Harrington et al., 2004; O’Leary et al., 2012), con- and heterospecific adult invertebrates (Osman & Whitlatch, 1995), predators and grazers (Connell & Anderson, 1999; Glynn, 1990; Lewis & Anderson, 2012), or changing environmental conditions that provide competitive advantages to certain species (Hallock & Schlager, 1986).

Eutrophication, the increase in nutrient availability influences biofilm diversity and composition (Kriwy & Uthicke, 2011; Webster et al., 2004; Witt, Wild & Uthicke, 2012a; Witt, Wild & Uthicke, 2012b). Further, eutrophication and overfishing (of herbivores) can also increase growth of benthic macroalgae such as filamentous algae (Jessen et al., 2013a), thereby providing the faster growing algae with a competition advantage over invertebrates, allowing them to take over suitable substrata. In contrast, some slow growing algae such as crustose coralline algae (CCA), important for coral recruitment (Harrington et al., 2004; Heyward & Negri, 1999), can be suppressed through reduced grazing (Jessen et al., 2013a). Additionally, the increase of certain filter feeders was linked to eutrophication and concomitant increase in organic matter in the water column that made them able to outcompete and prevent settlement of adjacent organisms (Chadwick & Morrow, 2011; Hallock & Schlager, 1986).

Further, overfishing can influence trophic interactions in two ways. Reducing the number of herbivores and invertebrate predators and therefore freeing macroalgae and certain invertebrates of their top-down control (Birkeland, 1977; Birrell, McCook & Willis, 2005; Diaz-Pulido et al., 2010; Osman & Whitlatch, 1995; Vine, 1974); similarly the reduction of predators can result in the release of top-down control of invertebrate feeders such as sea urchins (Hay, 1984; McClanahan & Shafir, 1990). As a consequence, the amount of invertebrate settlement can be strongly reduced (Myers et al., 2007), sometimes even down to almost zero (Vine & Bailey-Brock, 1984). Overfishing can furthermore lead to increased bioerosion rates (Tribollet & Golubic, 2011) that reduce suitable settlement habitat for new invertebrate settlement.

Although the top-down and bottom-up effects of overfishing and eutrophication have been intensively studied for benthic reef algal growth and development (e.g., Burkepile & Hay, 2006; Smith, Hunter & Smith, 2010; Jessen et al., 2013a), there are few studies that investigate the individual or combined impact on tropical sessile invertebrate settlement in this context. Only Tomascik (1991) and Hunte & Wittenberg (1992) looked at coral settlement patterns along an eutrophication gradient, although it is not clear if the observed influence was due to altered larval supply. Additionally, our understanding of the ecology of coral reefs in the Red Sea is largely focused on studies conducted in the Gulf of Aqaba, but not in the remaining Red Sea (Berumen et al., 2013).

Over 4 months this study simulated (a) overfishing by excluding larger predators and herbivores through cages and (b) eutrophication through the deployment of nutrient sources in an offshore reef in the Central Red Sea. We wanted to answer the question of how the individual and combined effects of overfishing and eutrophication impact the settlement of main sessile invertebrate groups.

Materials & Methods

Study site

The study was carried out over 16 weeks from June to September 2011 at the patch reef Al-Fahal that lies about 13 km off the Saudi Arabian coast in the Central Red Sea (N22.18.333, E38.57.768; see Jessen et al., 2012 for a map of the location). We selected this reef because of its relatively large distance from shore and presumably low impacts from potential fishing and land-derived nutrient import. The reef is characterized by high herbivore fish (22 g m−2) and sea urchin biomass (38 g m−2), low ambient concentrations of dissolved inorganic nitrogen (DIN = NH4+ + NO3− + NO2−; 0.9–1.8 µmol L−1), soluble reactive phosphorous (SRP = PO43−; 0.06–0.10 µmol L−1), dissolved organic carbon (DOC; 55–67 µmol L−1), and relatively high live coral cover (49% hard and soft coral cover; for full results see Jessen et al., 2013a).

Experimental setup

Ten terracotta tiles (plus two spare tiles) each 10 × 10 cm (100 cm2) were mounted on stainless steel screws at an angle of 45 degrees on each of 16 polyvinyl chloride (PVC) frames (50 × 75 cm; in total 160 tiles) approximately 10 cm above the reef substrate at 5–6 m water depths (Fig. 1) and accessible to invertebrate herbivores (C Jessen, pers. obs., 2011). Tiles were installed in 2 rows with a distance between 3 and 50 cm (Fig. 1). PVC frames were separated by 2–5 m. Prior to the start of the experiment, tiles were autoclaved to remove any interfering compounds that could have accumulated during tile production. Tiles were installed pairwise on top of each other with unglazed sides facing outside, resulting in an upper (light exposed) and lower (shaded) tile (Fig. 1). We applied four different treatments to the frames (each with n = 4): (1) control (only the equipped frame), (2) fertilizer (see nutrient enrichment section), (3) cage (hemispherical zinc galvanized cages with a mesh size of 4 cm and a diameter of 100 cm), and (4) a combination of cage and fertilizer tubes.

Figure 1 Schematic drawing of the experimental setup.

Shown is a PVC frame equipped with twelve tiles (two of them spare tiles) half of them in light exposed conditions and half of them in light shaded conditions. The dotted line shows where the glazed side of the tile (inward pointing side) has been pre-scored to facilitate tile division upon sampling.

The cages served to exclude larger predators and herbivores, as overfishing is rather affecting larger species. Smaller fish (small damselfish, parrotfish, wrasses and surgeonfish) were still able to access the insides of the cages. Cage controls were not used, since studies showed that similar cages even with a lower mesh size did not affect water movement, light availability, and sedimentation rates (Burkepile & Hay, 2007; Miller et al., 1999; Smith, Smith & Hunter, 2001).

Eutrophication was simulated by deploying four fertilizer tubes around the frame, consisting of perforated PVC tubes filled with approximately 580 g Osmocote fertilizer (Scotts, 15% total nitrogen (in form of nitrate & ammonium), 9% phosphate (phosphoric pentoxide), and 12% potassium oxide) embedded in 3% agarose. Fertilizer was deployed once without replenishments, but regular monitoring of nutrient concentrations assured continuous release rates.

Treatments were not randomly assigned to the frames, instead the sequence control, fertilizer, cage, combination was repeated four times along the reef to control for potential biases such as microhabitats.

One pair of tiles (light-exposed and shaded) per frame was collected after 1, 2, 4, 8, and 16 wk(s) using SCUBA. To facilitate tile division under water tiles were pre-scored (notched) on their glazed side with the help of a tile cutter before the start of the experiments (Fig. 1). Upon sampling, tiles were divided in half (each 50 cm2; an area which had been shown to be large enough from asymptotes of species-area curves by Hixon & Brostoff, 1996) and then wrapped separately in ziplock bags. They were brought on board within 30 min and half of them was immediately flash frozen in liquid nitrogen for subsequent microbial analyses (results reported elsewhere), while the other half was handled as described below.

To test the success of fertilization, water samples (5 L) were taken directly before collecting tiles at each time point with large ziplock bags directly from above each frame (in total n = 40 for nutrient enriched as well as non-enriched samples). From this stock 50 mL were filtered on pre-combusted Whatman-GF/F filters and used for inorganic nutrient measurements. The analyses of DIN and SRP were performed using continuous flow analyzer (FlowSys Alliance Instruments).

Invertebrate identification and enumeration

In order to remove attached sediment, precipitates, and mobile invertebrates, light-exposed and shaded tiles were rinsed with fresh water. Invertebrate classification was conducted with a dissection microscope (Zeiss Stemi 2000; 7.7-fold magnification). All sessile invertebrates visible under the dissection microscope were identified with the help of Vine (1986) and grouped to the following easily distinguishable categories: Scleractinia (Cnidaria), Bivalvia (Mollusca), Bryozoa, and Polychaeta (Annelidae). We counted single animals (Scleractinia, Bivalvia, and polychaetes, such as Spirorbis sp. or Pomatoceros sp.) or colonies (Bryozoa and other polychaetes, such as Filograna sp.) on each tile to quantify the number of individual settlement events. It is likely that factors other than settlement such as competition, predation, and overgrowth affected the number of organisms in the course of the study. However, by considering only sessile and calcareous organisms and thoroughly searching the surface using a dissection microscope, we tried to minimize potential biases as much as possible. Nevertheless, numbers can be slightly underestimated since we cannot rule out that settlers arrived but did not persist.

Algal composition and algal biomass (only light-exposed tiles) was determined in the laboratory after invertebrate counting by taking pictures of submerged tiles and analyzing them using 100 randomly overlaid points using Coral Point Count with Excel extensions (CPCe) 4.1 (Kohler & Gill, 2006). Primary algal groups were filamentous algae and non-coralline crusts on light-exposed tiles and crustose coralline algae (CCA) and non-coralline red crusts (such as Peyssonnelia) on shaded tiles. Foliose macroalgae such as Padina, Lobophora, or Halimeda were not found. See Jessen et al. (2013a) for full results of algal cover.

Data of 1 of 16 frames (No. 4, combined treatment) was removed from the dataset, as cage pictures and tile appearance indicated access of large predators and herbivores to this setup.

Statistical data analysis

T-tests were used for analyzing inorganic nutrient concentrations at each sampling point. To meet assumptions of normal distribution DIN-data were inverse square root (1/sqrt(x)) transformed. All invertebrate groups were tested for the individual and interactive effects of cage, fertilizer, and time with a 3-factorial generalized linear model (GLM) in R (R Development Core Team, 2012). To cope with over- and underdispersion we used either quasi-GLM models (hard corals, polychaetes) or negative binomial model (Bivalvia, Bryozoa), depending which model fit the data better based on pseudo-R2 scores (Zuur et al., 2009). For comparison of the different treatments, we used Tukey post hoc tests (‘glht’ function) of the ‘multcomp’ package.

Results

The simulation of eutrophication constantly and significantly increased SRP concentrations compared to the controls (Fig. S1). DIN concentrations were also constantly increased, but did not always significantly differ from the controls (Fig. S1). Both, ambient and enriched treatments experienced a peak in DIN concentrations after 4 weeks.

Over the sampling period of 16 weeks, 99.9% of all observed sessile invertebrates settled on the shaded tiles. The exceptions were 1 hard coral recruit (control 2 wks), 5 polychaetes (1 × control 2 wks; 3 × fertilizer 4 wks; 1 × combined 8 wks), and 2 bryozoan colonies (cage 16 wks). Because of this one-sided distribution, the following results stem exclusively from invertebrate observations of the shaded tiles (total 6,862 counts, and an average of 91 counts per shaded tile). Figure 3 shows representative photographs of light-shaded tiles after 16 weeks of deployment in the reef.

On a temporal scale, polychaetes occurred first after 1 week, bryozoans after 2 weeks, hard corals after 4 weeks, and bivalves after 8 weeks, however, there was no treatment-specific pattern when first settlement occurred (Figs. 2B, 2D, 2F and 2H). Other potential sessile invertebrate groups such as sponges, soft corals, crustaceans, and ascidians were not observed on the analyzed tiles, however the latter group appeared once on a spare tile after 16 weeks.

Figure 2 Invertebrate settlement numbers (depicted per cm−2; mean ± SE) on shaded tiles.

Left column (A, C, E, G) shows settlement numbers per treatment averaged over all tiles and right column (B, D, F, H) shows temporal development of counted recruits of all 4 treatments. p-values were calculated from a 3-factorial GLM and originate from analysis across the whole study period (see Table S1 for full results). Dashes represent factors that have been excluded by the model reduction. Abbreviations: C, Cage; F, Fertilizer; T, Time. Treatments with same small letters are not significantly different (p > 0.05) in post hoc pairwise comparisons.

Figure 3 Representative photographs of light-shaded tiles after 16 weeks of deployment in the reef.

White bars in the central upper right area of each picture are reflections caused by a camera flash. Hemi-circle holes at the central lower edge were used for screws to attach tiles. According to a point count analysis ran in Coral Point Count with Excel extensions (CPC), average invertebrate cover did not exceed 8%. See Jessen et al. (2013a) for full results.

On the controls, all observed invertebrate groups were present at their lowest abundance compared to the other treatments, except hard coral settlement which was highest in this treatment (Fig. 2A).

Simulated overfishing reduced hard coral numbers to zero (Fig. 2A), but significantly increased settlement of polychaetes (Fig. 2G). However, simulated overfishing did not show any significant effects on settlement of bryozoans and bivalves (Figs. 2C and 2E).

Under simulated eutrophication, hard coral settlement was significantly decreased by 11-fold relative to controls (Fig. 2A), while bryozoans, bivalves, and polychaetes were not significantly affected by this treatment (Figs. 2C–2H).

The combination of manipulated eutrophication and overfishing significantly increased settlement of bryozoans and bivalves 7 and 11-fold relative to controls (Figs. 2C and 2E). Relative to simulated overfishing, the combined treatment significantly increased settlement of bryozoans 4-fold and that of bivalves 11-fold, but decreased settlement of polychaetes 2-fold, while settlement of hard corals was not affected. Relative to simulated eutrophication, the combined treatment significantly increased settlement of bryozoans 3-fold and bivalves 7-fold, while settlement of hard corals and polychaetes was not affected.

Except for bryozoans, all other groups showed significant interaction effects, i.e., their response to one manipulated factor depended on the level of the other factor (Fig. 2, Table S1).

Discussion

Simulated overfishing increased settlement of polychaetes compared to controls. These observations are concordant with Vine (1974), who observed increased spirorbid settlement in caged treatments. Interestingly, the positive effect of simulated overfishing on settlement of polychaetes was not visible in the combined treatment with increased nutrient availability. A possible explanation could be the presence of heterospecific invertebrates (i.e., bryozoans, bivalves) that can suppress settlement in their vicinity (Osman & Whitlatch, 1995). This hypothesis is supported by the fact that the different polychaete settlement responses between simulated overfishing and combined treatments were not visible before the occurrence of bryozoans and bivalves, which started after 8 weeks.

Simulated eutrophication alone only caused decrease of hard coral settlement, while all other invertebrates were neither positively nor negatively affected by this treatment. This finding is confirmed by the studies of Tomascik (1991) and Hunte & Wittenberg (1992), who also observed less hard coral settlement in eutrophic reefs and suggest that eutrophic conditions may alter the complex set of physical, chemical and/or biological signals that trigger settlement of coral larvae. However, it is not clear if such differences were caused by negative settlement behavior, post-settlement mortality, or reduced larval supply (i.e., reduced coral fecundity) as observed by Loya et al. (2004) as a response to eutrophication. Large differences in functional algal cover between simulated eutrophication and control treatments did not exist (Jessen et al., 2013a). However, algae species were not identified on the species level, but potential differences on that level therefore may have occurred and influenced the settlement as shown for coralline algae by Harrington et al. (2004). Furthermore, as shown for coral fragments in a parallel experiment (Jessen et al., 2013b), increased nutrient concentrations may have altered the microbial community structure of biofilms, thereby changing chemical and structural cues that influence settlement.

The combination of manipulated overfishing and eutrophication resulted in the highest settlement numbers of bivalves and bryozoans, that were both significantly increased compared to manipulated overfishing and eutrophication treatments. However, algal cover, an important settlement cue, did not substantially vary between combined and simulated overfishing treatments (Jessen et al., 2013a). We propose therefore that the observed differences were due to (a) indirect interaction effects of predator/herbivore exclusion, (b) differences in bacterial and diatom biofilm composition (Dahms, Dobretsov & Qian, 2004; Yang et al., 2013) and (c) effects of microalgae benefiting from increased nutrients (Posey et al., 2002).

In this study, sessile invertebrates settled almost exclusively on shaded, compared to light-exposed tiles. This light exposure-specific pattern has been confirmed for corals by studies from other reefs (Birkeland, 1977; Harrison & Wallace, 1990; Sawall et al., 2013), and contrasts the presence of algae biomass and abundance of filamentous algae that was highest on light-exposed tiles during the present study (Jessen et al., 2013a). While these filamentous algae can prevent invertebrate settlement (Arnold, Steneck & Mumby, 2010; Glasby & Connell, 2001; Virgilio, Airoldi & Abbiati, 2006), encrusting algae, i.e., CCAs, often facilitate and induce invertebrate settlement (Arnold, Steneck & Mumby, 2010; Heyward & Negri, 1999; Morse et al., 1996; Negri et al., 2001; Whalan, Webster & Negri, 2012). Correspondingly, encrusting algae were not observed on the light-exposed tiles, but were abundant on the shaded tiles, particularly in non-caged treatments (Jessen et al., 2013a). Nevertheless, invertebrates were obviously present on light-exposed substrate in natural reefs. It may be that adequate settlement substratum for CCA exhibit delayed growth on light-exposed underground (Smith, Hunter & Smith, 2010) and thereby delaying sessile invertebrate settlement. This suggests the need for studies over longer time spans to study invertebrate settlement on light-exposed substrata. While other invertebrate groups that are typically associated with coral reefs including sponges, soft corals, crustaceans, and ascidians were absent in this experiment, they were found in other, though longer lasting, similar experiments (e.g., Sawall et al., 2013). Their lack in this study may be either explained by the absence of reproduction events during the study period or delayed settlement on artificial substrata as suggested by the observation of ascidians on a spare tile after 16 weeks.

The absence of all hard coral settlement in the simulated overfishing treatments may be caused by the presence of more competitive invertebrates that prevented settlement or covered corals (Birkeland, 1977; Sawall et al., 2013), filamentous algae (Arnold, Steneck & Mumby, 2010; Birrell, McCook & Willis, 2005; Kuffner et al., 2006), as well as the lower abundance of coralline algae (O’Leary et al., 2012), as these factors were significantly influenced by simulated overfishing on the same tiles (Jessen et al., 2013a).

In a recent review, Cooper, Gilmour & Fabricius (2009) summarized and evaluated potential bioindicators for coral reef health and water quality, ranging from species presence and composition to physiological and isotopic parameters. Although their review included coral recruitment, other sessile invertebrates were not considered. The findings of the present study suggest settlement of coral reef associated sessile invertebrates as specific bioindicators for overfishing and a combination of that with eutrophication. For overfishing, this may be an increase in polychaete settlement and a decrease for that of hard corals. For eutrophication the sole decrease of hard coral settlement, and for a combination of both stressors this may be an increase in bryozoan and bivalve settlement. Advantages of this approach would be the cost-effective and relative easy measurement together with low systematic knowledge that is needed to identify the taxonomic groups.

As settlement is only one process in the successful recruitment of an organism, it could be interesting to compare the profiteers of the different treatments in this study with known juvenile mortality rates. This would allow improving predictions on potential sessile invertebrate outbreaks or phase shifts.

Studies showed that hard coral settlement experiences up to 100% mortality within the first year with regional differences: e.g., in French Polynesia 19–56% over 14 months (Penin et al., 2010), in Jamaica 91–95% over 9–10 months (Rylaarsdam, 1983), in Florida (annual) 22–49% (Miller, Weil & Szmant, 2000), in Bonaire 32% over 6 months (Bak & Engel, 1979), in the Great Barrier Reef (annual) 36% (Connell, 1973), 90% over 4 months (Harriott, 1983), 67–86% over 8–9 months (Babcock, 1985), and 96–99% over 4 months (Babcock & Mundy, 1996). The outcomes from non-coral invertebrates from other ecosystems were similar with many studies reporting mortalities of >90% (reviewed in Gosselin & Qian, 1997).

To conclude, although the reef appears to be in healthy condition, non-coral invertebrates such as polychaetes or bivalves and bryozoans may rival hard coral dominance at shaded reef locations if simulated threats are not controlled in the study area. This can lead to phase-shifts, potential alternative stable states that may impact the ecology of coral reefs (Norström et al., 2009).

Supplemental information

Figure S1 Inorganic nutrient concentrations

Dissolved inorganic nitrogen (DIN) and soluble reactive phosphate (SRP) concentrations (μ mol L−1; means ±SE) in the nutrient enrichment treatments (fertilizer & combined) and the non-enriched treatments (control & cage). Small letters (a—SRP; b—DIN) indicate statistical significant differences of p < 0.05 (t-test).

Click here for additional data file.

Table S1 Results of the 3-factorial GLM of invertebrate groups

Abbreviations: Cage (C), Fertilizer (F), and Time (T). Significant results are indicated in bold by asterisks. P-values of 0.000 represent values < 0.001. Dashes represent factors that have been excluded by the model reduction.

Click here for additional data file.

Many thanks for support go to the CMOR team at KAUST, L Smith, F Mallon, E Aravantinos, and the boat crew K Al-Moullad, E Al-Jahdali, G Al-Jahdali. We also would like to acknowledge the field and laboratory assistance of C Roder, JF Villa Lizcano, A Roik, C Arif, M Kruse, and C Jhen. For invaluable advice and support in the laboratory we thank N Fayad, M Masry, D Dassbach, and M Birkicht. Furthermore, the authors want to thank the editor, Vera Vasas, Ian Hendy, and an anonymous reviewer who provided valuable comments to improve the manuscript.

Additional Information and Declarations

Competing Interests

Author Contributions

The authors declare there are no competing interests.

Christian Jessen conceived and designed the experiments, performed the experiments, analyzed the data, wrote the paper, prepared figures and/or tables, reviewed drafts of the paper.

Christian R. Voolstra conceived and designed the experiments, performed the experiments, contributed reagents/materials/analysis tools, wrote the paper, reviewed drafts of the paper.

Christian Wild conceived and designed the experiments, wrote the paper, reviewed drafts of the paper.

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
