# Peer review of "In situ effects of simulated overfishing and eutrophication on settlement of benthic coral reef invertebrates in the Central Red Sea"

_PeerJ, doi:10.7717/peerj.339_

## Round 0.1 · original submission · Minor Revisions

All reviewers deemed the work of publishable quality, however, the reviewers have outlined a number of corrections to your manuscript which may aid the overall understanding of the experimental procedure and add context to the experimental design in relation to overfishing/eutrophication.

·

Basic reporting

In general, the manuscript is sound. However, the author(s) have made quite a few very minor errors - that must be altered before submission. I have attached a PDF with my explicit details.

Experimental design

No Comments

Validity of the findings

No Comments

Additional comments

I am very much in favour of applied and experimental ecological publications – and in my view, this work adds to current coral-reef research. The work is innovative, engaging and insightful. Furthermore, it addresses prominent issues related to current exploitation of ecosystem resources. I do, however, have one ‘gripe’. I would suggest that it may not be immediately obvious to non-marine ecologists that the caged tile samples signify ‘over-fishing’. The manuscript may benefit by adding a few sentences on how you classify or deem ‘over-fishing’. In my opinion, this paper is of a quality that should be published.

Reviewer 2 ·

Basic reporting

Overall the manuscript is well written although it does contain various sections which need to be revised to be understandable. There also are a large number of sentences that do require a Reference which the authors seemed to have failed to insert. Similarly the reference list is missing completely and needs to be added to the ms. Some relevant figures and data is still missing from the paper. The nutrient measurements obtained need to be added to the paper.
The ms does not report on the full data of the experiment described. Only data of invertebrate recruitment is provided for the shaded tiles. Results of the algae cover of the light exposed settlement tiles are not provided but are used in the discussion. A non-traceable reference is given for this data. I find that this lies in conflict to the required self-contained nature of papers published in PeerJ. The authors should add this information to the paper as it is part of the same experiment and also the data forms part of the discussion and explanation of observed trends.

Experimental design

Overall the design and analysis seems fine but I would include a schematic drawing of the design. Some more details of the statistical outputs would have been advisable. The paper does not follow the usual convention of reporting statistical results.

Validity of the findings

Overall the study is sound but I have been struggling with the interpretation of some of the results see the two points below. Also I think the authors need to further highlight that results obtained may be only specific to the study site and may not be able to be extrapolated to other areas. Also as not all data of the experiment is provided I feel that I only got half the story and are unable to evaluate the study in its full context
Line 192-196 This justification does not make sense to me. There is something strange here. Please rethink and reformulate.
Line 127-131 it is impossible to control these processes in such experiments and they are just part of such experiments. Suggesting that by careful handling this would be improved is not possible. The authors should just acknowledge the limitation no more is needed. As such the authors do report on settlement and post settlement processes that are occurred within the 4 month experiment.

Additional comments

Line 24 please add ……simulated overfishing through predator exclusion prevented
Line 54 A sentence should not start with an Or change to …. Similarly the reduction of predators can result……
Line 57 Overfishing can furthermore lead to
Line 63-65 This sentence is not understandable please reformulate and add the appropriate references.
Line 67 simulated overfishing through predator exclusion and………
Line 74 please provide coordinates in Degrees and Minutes
Line 80-111 please provide a schematic drawing of the experimental setup.
Line 95 provide reference for this statement
Line 106 please explain the term pre-scored?
Line 109 replace where with and
Line 122-126 It is not clear to what taxonomic level the species were identified. The analysis was done on group level. I still feel it would have been useful to report on species or genus level at least within a table.
Line 127-131 it is impossible to control these processes in such experiments and they are just part of such experiments. Suggesting that by careful handling this would be improved is not possible. The authors should just acknowledge the limitation no more is needed. As such the authors do report on settlement and post settlement processes that are occurred within the 4 month experiment.
Line 132-138 the algae cover data should have been presented here in the same paper. I do not see any real reason for separating the two datasets.
Line 142 unable to locate the results in the ms???
Line 147 different distributions do not correct for over and or under dispersion they are able to accommodate different distributions.
Line 152 Figure is missing
Line 157 SRP and DIN should be written out at the beginning of the discussion
Line 189 Reference missing
Line 192-196 This justification does not make sense to me. There is something strange here. Please rethink and reformulate.
Line 199 Reference missing
Line 203 check sentences
Line 207 Reference missing
Line 212-216 data not shown here, see arguments above
Line 221 CCA please write out in full before abbreviating

·

Basic reporting

The manuscript focuses on the effects of eutrophication and overfishing on the settlement of reef-associated invertebrates. The topic is addressed appropriately, and it is self-contained enough to stand as a ‘unit of publication’. However, settlement at the colonization phase is only one of the factors that influence coral reef dynamics, and the manuscript in its current form is missing a discussion on the long-term effects initial settlement is expected to have on the community composition.
The manuscript is clearly written and easy to read. The figures are chosen appropriately. For clarity, I would suggest the authors to consider the inclusion of an additional figure that summarizes the mechanisms by which eutrophication and overfishing is expected to alter settlement.

Experimental design

No comments.

Validity of the findings

The statistical analysis contains a mistake for the coral recruits: if the fertilizer treatment is not expected to have any effect then the combined fertilizer x cage interaction cannot be considered.
Line 229 and then on. I think the result that algae dominated all light-exposed tiles should be further elaborated upon. Under what light conditions do corals naturally occur? As you mention, invertebrates are present at light-exposed substrate in natural reefs. If, on light exposed tiles, initial settlement of algae is expected to give way to invertebrates on the long term, what is the justification for studying short-term settlement on shaded tiles?

Additional comments

Did the tiles become fully covered with settlers during the experiment? Please report on this. Fully covered tiles would indicate that competition and persistence is expected to come into play.
Line 177. “Zero relative to controls” – change to “zero”.
Line 202. This argument is not clear to me. If conspecific polychaetes suppress settlement, how is this behaviour modified in the cage but not in the combined treatment? Couldn’t it be that increased polychaete settlement is indeed the result of predator exclusion, but this effect is offset in nutritionally enhanced conditions by competition with bivalves and bryozoans?

---

## Round 0.2 · accepted · Accept

(No comments from editor)